# Application of ZnO Nanocrystals as a Surface-Enhancer FTIR for Glyphosate Detection

**DOI:** 10.3390/nano11020509

**Published:** 2021-02-17

**Authors:** Anderson L. Valle, Anielle C. A. Silva, Noelio O. Dantas, Robinson Sabino-Silva, Francielli C. C. Melo, Cleumar S. Moreira, Guedmiller S. Oliveira, Luciano P. Rodrigues, Luiz R. Goulart

**Affiliations:** 1Nanobiotechnology Laboratory, Institute of Genetics and Biochemistry, Federal University of Uberlândia, Uberlândia 38402-022, MG, Brazil; andersonluis.valle@gmail.com (A.L.V.); francielli.melo@gmail.com (F.C.C.M.); 2Laboratory of New Insulating and Semiconductors Materials, Institute of Physics, Federal University of Uberlândia, Uberlândia 38408-100, MG, Brazil; noelio@fis.ufal.br; 3Laboratory of New Nanostructured and Functional Materials, Institute of Physics, Federal University of Alagoas, Maceió 57072-970, AL, Brazil; 4Institute of Biomedical Sciences, Federal University of Uberlândia, Uberlândia 38402-022, MG, Brazil; robinsonsabino@gmail.com; 5Electrical Engineering Department, Federal Institute of Paraíba, João Pessoa 58015-020, PB, Brazil; cleumar.moreira@ifpb.edu.br; 6Institute of Chemistry, Federal University of Uberlândia, Uberlândia 38402-022, MG, Brazil; guedmiller@ufu.br; 7Institute of Engineering, Science and Technology, Federal University of Jequitinhonha and Mucuri’s Valleys, Janaúba 39447-814, MG, Brazil; luciano.rodrigues@ufvjm.edu.br

**Keywords:** nanocrystals, nanocomposites, FTIR enhancement, FTIR spectroscopy, silver oxide, zinc oxide, Ag-doped ZnO, glyphosate

## Abstract

Glyphosate detection and quantification is still a challenge. After an extensive review of the literature, we observed that Fourier transform infrared spectroscopy (FTIR) had practically not yet been used for detection or quantification. The interaction between zinc oxide (ZnO), silver oxide (Ag_2_O), and Ag-doped ZnO nanocrystals (NCs), as well as that between nanocomposite (Ag-doped ZnO/AgO) and glyphosate was analyzed with FTIR to determine whether nanomaterials could be used as signal enhancers for glyphosates. The results were further supported with the use of atomic force microscopy (AFM) imaging. The glyphosate commercial solutions were intensified 10,000 times when incorporated the ZnO NCs. However, strong chemical interactions between Ag and glyphosate may suppress signaling, making FTIR identification difficult. In short, we have shown for the first time that ZnO NCs are exciting tools with the potential to be used as signal amplifiers of glyphosate, the use of which may be explored in terms of the detection of other molecules based on nanocrystal affinity.

## 1. Introduction

Glyphosate ((N-phosphonomethyl) Glycine) is the most commercialized herbicide worldwide, with the amount used globally reaching billions tons [1]. It has seen a dramatic increase in usage since genetically modified glyphosate-resistant crops were introduced late in the 20th century [2]. A commercial formulation of glyphosate (GLIFOTAL TR) consists of isopropylamine salt and the surfactant polyoxyethylene amine [3]. This herbicide was considered “toxicologically harmless” for animals and the environment [4,5,6] due to its ability to degrade in soil microbes and bind to soil colloids [7]. However, at present it is back in focus due to its carcinogenic effects (according to the International Agency for Research on Cancer) based on evidence from agriculture exposure and laboratory animal data [8].

As a matter of fact, there are more than eighty methods used to detect glyphosate [9]; however, its detection and quantification processes are generally expensive and slow, which means that governmental control measures are ineffective. European Union (EU) authorities have been regularly monitoring glyphosate levels in cereals since 2010. Still, the challenge in testing glyphosate residues on imported genetically modified soybeans (GMS) remains, and Brazil is one of the biggest producers of GMS. Even in Europe, only a small number of testing laboratories can detect glyphosate [10].

Glyphosate’s capacity to adsorb strongly onto clay minerals [11] and organic or mineral particles in water [12,13] and its high affinity with metal cations impose great difficulties on its detection without pretreatment methods [14]. The behavior of glyphosate has been examined by several research groups, showing the tendency of transitions in some trivalent metal ions and divalent alkaline-earth metal ions to form 1:1 (e.g., Ca(II) and Cu(II) [15,16]) and 2:1 metal chelates with glyphosate in solution [17]. In the commonly used density functional theory, molecular modeling methods show that zinc is the most stable element to form tetrahedral and octahedral complexes with glyphosate. Thus, the affinity dominances of various elements are as follows: Zn > Cu > Co > Fe > Cr > Al > Ca > Mg [18].

It is known that most agricultural soils contain Zn (10–300 mg kg^−1^) [19] and that an increase in glyphosate use has an effect on the availability of Zn on soils [20]. On the other hand, the interaction of glyphosate with Ag has barely been discussed and has generally been used only for biosensing purposes [21,22,23]. Zinc is usually found bonded with oxygen (ZnO), with the resulting compound widely used as an additive in several products and materials applied in foods and the degradation of pollutants [24]. Zinc Oxide (ZnO) nanoparticles can be amorphous or crystalline, the difference being that nanocrystals (NCs) are highly stable and do not present genotoxicity, unlike amorphous nanoparticles [25]. ZnO nanocrystals (NCs) are also used in electronic devices because they have supercapacitor properties [26].

Thus, in this work, the interaction of glyphosate molecules with Ag-doped ZnO, silver oxide (Ag_2_O) nanocrystals, and nanocomposite (Ag-doped ZnO/AgO) will be investigated, with implications possibly relevant to biosensing applications. These interactions were investigated by Fourier transform infrared spectroscopy (FTIR) and atomic force microscopy (AFM), and the results have shown for the first time that ZnO NCs can be used as FTIR enhancers, enabling glyphosate detection using several technologies.

## 2. Materials and Methods

### 2.1. Synthesis and Characterization of Nanocrystals

ZnO, Ag_2_O, and Ag-doped ZnO nanocrystals and a nanocomposite (Ag-doped ZnO/AgO) were synthesized by coprecipitation methodology patented according to process number BR 10 2018 007714 7-National Institute of Industrial Property in the absence of surfactants [27]. Structural properties were investigated using X-ray diffraction (XRD) (DRX-6000, Shimadzu, Kyogo, Japan) with monochromatic radiation Cu-K_α1_ (λ = 1.54056 Å). The AgO phase percentage was determined by the ratio of the integrated intensities of AgO/ZnO diffraction peaks from XRD results. We also used FTIR and AFM to characterize nanomaterials.

### 2.2. Characterization of GLIFOTAL TR with Nanocrystals and Nanocomposite

Samples were prepared from a 10^−2^ (*v*/*v*) dilution of GLIFOTAL TR (1:100 ultrapure water). Nanomaterials were also diluted in ultrapure water (10 mg mL^−1^), which was mixed with GLIFOTAL TR dilutions. The final solution was stirred manually at a medium speed. To prove interaction properties, NCs were also diluted in ultrapure water (1.5 mg mL^−1^).

### 2.3. Fourier Transform Infrared Spectroscopy (FTIR)

Samples’ spectra were recorded in a 4000–400 cm^−1^ range using an FTIR spectrophotometer (Vertex 70, Bruker Optik, Ettlingen, Germany) with a microattenuated total reflectance (ATR) accessory. The crystal material in the ATR unit was a diamond disc used as an internal reflection element. The sample penetration depth ranged between 0.1 and 2 μm. The samples were dropped 2 µL twice and dried using a triple dental syringe to obtain FTIR spectra at room temperature. The air spectrum was used as a background in the FTIR analysis. The sample spectra analyses were obtained with 2 cm^−1^ of resolution and 34 scans.

### 2.4. Atomic Force Microscopy (AFM)

AFM imaging was performed to view the interaction of GLIFOTAL with NCs and nanocomposite. For evaluation, we dropped 3 µL of GLIFOTAL TR NCs 10^−4^ (*v*/*v*) solution onto a mica sheet surface and submitted it to an AFM with a high resolution scanning probe [28]. Mica sheets and the nanofilm formed by GLIFOTAL TR when the sheets were overlapped were used as control groups.

### 2.5. Enhancement Properties Analysis

FTIR spectra were obtained in order to investigate glyphosate’s interactions with its commercial formulation (GLIFOTAL TR), nanocrystals (ZnO, Ag-doped ZnO, and Ag_2_O), and nanocomposite (Ag-doped ZnO/AgO) at four dilutions in 10^−2^ (*v/v*) (usually used for soil applications), 10^−4^ (*v*/*v*), 10^−6^ (*v*/*v*), and 10^−8^ (*v*/*v*). GLIFOTAL TR is formed by Isopropylammonium N (phosphonomethyl)-Glycine at 480 g L^−1^ (48% *m*/*v*), with the acid equivalent at 360 g L^−1^ (36% *m*/*v*) and the other ingredients at 673.4 g L^−1^ (67.34% *m*/*v*). 

## 3. Results

X-ray diffraction was used to investigate the structural properties of the nanocrystals and nanocomposite. Figure 1 shows the XRD patterns of nanomaterials at room temperature. In Figure 1A, observed that the diffraction patterns are characteristic of ZnO wurtzite crystal structure (JCPDS card No. 36-1451). The silver oxide standards correspond to the cubic crystalline structure of Ag_2_O nanocrystals (JCPDS 76-1393).

In order to investigate the effect of the incorporation of Ag into ZnO, zooming in on the diffraction peak of (100) shows smaller angular shifts relative to the ZnO in nanocomposite and Ag-doped ZnO, as shown in Figure 1B. This result confirms silver’s substitution of zinc into ZnO’s crystalline structure, since Ag^+2^ has an ionic radius (1.26 Å) larger than that of Zn^+2^ (0.74 Å). The magnification also shows a typical AgO peak (JCPDS no: 43-1038) in the nanocomposite (blue line). The AgO phase percentage estimated in the nanocomposite is 22.4%. The nanocomposite formed consists of Ag-doped ZnO and AgO NCs (Ag-doped ZnO/AgO) [27].

In order to investigate the interactions of nanomaterials with glyphosate functional groups, the infrared spectroscopy technique was used. The nanomaterials were mixed with GLIFOTAL TR in ultrapure water at three dilutions: 10^−2^ (*v*/*v*) (which is usually used for soil applications), 10^−4^ (*v*/*v*), and 10^−6^ (*v*/*v*). Figure 2A shows the FTIR spectra of the nanomaterials studied. The characteristic bands were represented between 400–600 cm^−1^ related to Zn-O and Ag-O bonds [29,30]. Figure 2B observes the spectra of GLIFOTAL TR (10^−2^, 10^−4^, and 10^−6^ (*v*/*v*)) and GLIFOTAL TR Ag_2_O NCs in 10^−2^ (*v*/*v*) concentrations. These bands of nanomaterials are low in intensity compared to the GLIFOTAL TR bands. This analysis also shows that even a low Ag_2_O NCs concentration can reduce the FTIR vibrational signal.

Figure 3A shows the FTIR spectra of GLIFOTAL TR in three different concentrations (10^−2^, 10^−4^, and 10^−6^ (*v*/*v*)) and the GLIFOTAL TR nanomaterials at a concentration of 10^−6^ (*v*/*v*). Figure 3B shows the FTIR spectra of glyphosate in various concentrations and mixed with nanomaterials at 10^−6^.

The enhancement property was observed when ZnO NCs were added to the GLIFOTAL TR solutions diluted at 10^−6^ (*v*/*v*) (480 ng mL^−1^), at which point the solution’s bands were more intense than those of solutions diluted at 10^−2^ (*v*/*v*) without NCs. On the other hand, with the addition of Ag-doped ZnO, a slight suppression of the glyphosate bands was observed, decreasing even more when the nanocomposite was incorporated. That being said, all of the glyphosate’s vibration modes were observed. Therefore, ZnO NCs are exciting tools with the potential to be used as signal amplifiers of GLIFOTAL TR.

Figure 4 shows the vibration modes of glyphosate’s molecular group. It observes that their intensity changes when they interact with each sample, considering the most and least diluted concentrations of GLIFOTAL TR (10^−6^ and 10^−2^ (*v*/*v*)), respectively. Each bond that composes each glyphosate functional group’s molecular structure may present higher or lower energy after an interaction with the samples. Each color represents a specific vibration mode; blue represents the highest intensity, while red represents the lowest intensity.

In Figure 4A, *R* is the Pearson’s correlation among the samples and their respective glyphosate vibrational mode intensity; the closer the correlation is to 1, the more significant the positive correlation is. The first three columns, composed of GLIFOTAL TR in different concentrations, show the analytical limitations of the technique without nanomaterials, since just 10^−2^ (*v*/*v*) dilution could be observed (shown by more intense coloring). In this group, GLIFOTAL TR (10^−2^) shows high-intensity bands at 1568, 1557, 1170, 1081, and 1031 cm^−1^. The nanomaterials’ interaction with the GLIFOTAL TR solution shows that, while ZnO NCs enhanced the herbicide’s bands, the nanomaterials with Ag diminished it. To understand where the nanocrystals interacted with the glyphosate structure, the intensities of the FTIR bands were calculated for each vibrational mode and normalized by the lowest dilution. Thus, the ratio’s columns in Figure 4B represent the values of the nanomaterials’ columns divided by the values of the GLIFOTAL TR (10^−2^) column.

AFM images of the samples are shown in Figure 5. The AFM images for the control group show a smooth surface and roughness for GLIFOTAL TR. The right panels show the two-dimensional AFM images. ZnO NCs in water were commonly found ungrouped, dispersed on water, and in association with GLIFOTAL TR. This specific interaction is evidenced by a single nonclustered unit. Alternatively, the samples containing Ag are always seen in clusters in water when associated with GLIFOTAL TR. In water, it was observed that nanomaterials presented different degrees of aggregation. While ZnO and Ag-doped ZnO NCs were well dispersed, Ag_2_O NCs and nanocomposite were not.

In the AFM images of nanocrystal/GLIFOTAL TR groups, different aggregation degrees were observed when nanomaterials were associated with GLIFOTAL TR, and these were more intense for samples containing Ag (Ag-doped ZnO, nanocomposite, and Ag_2_O). Thus, nanomaterials’ interactions with Ag (Ag-doped ZnO, nanocomposite, and Ag_2_O) and glyphosate are relatively more intense, facilitating aggregation.

## 4. Discussion

Glyphosate detection and quantification is expensive and slow [31,32]. Governmental control of the process is ineffective because multiresidue methods cannot detect glyphosate. The impact of this knowledge gap on the health system and the public economy is unknown. Hence, the concept of “glyphosate’s paradox” was raised, which means that despite glyphosate being the most widely used agrochemical globally, it is also the least analytically well-determined agrochemical [7,33].

The regulatory rules regarding the maximum residue limit (MRL) for both crops and water adopted by each country differ significantly. For example, the value established by the United States Environmental Protection Agency (EPA) for soybeans was 20 mg kg^−1^. Independent of the compound’s structure or activity, the European Union has set the MRL of pesticides at 0.1 ng mL^−1^, while the EPA’s MRL, established in terms of the persistence and toxicity of each pesticide individually, was 700 ng mL^−1^ [34]. The Canadian Drinking Water Guideline recommends an MRL of 280 ng mL^−1^, and the Brazilian Surveillance Agency (ANVISA) recommends an MRL of 500 ng mL^−1^. It is important to emphasize that analyses on current approvals by the EU and USA regarding glyphosate levels suggest that the established acceptable daily intake levels are inaccurate and dangerously high [35], mainly because agencies have used information studies performed by industries benefitting from these high intake levels. These facts have raised questions about how safe high glyphosate levels are, and these questions are further complicated by the fact that many technical approaches present limits of detection (LODs) which are low compared to which each agency would be interested in controlling.

In this way, we have shown in this investigation that ZnO NCs may be used as an enhancing agent for the FTIR glyphosate detection methodology for the first time. The opposite effect was observed with Ag-doped ZnO NCs and nanocomposite (Ag-doped ZnO/AgO), since the Ag ion on the other hand, acted as a chelating agent in glyphosate, leading to full signal suppression. Thus, interestingly, we have also shown for the first time that nanocomposite mixed into water has almost no effect on FTIR, suggesting that the ZnO NCs did impact the glyphosate structure and induce a differential vibrational intensity.

It is known that ZnO interacts with glyphosate and has a direct effect on the environment, impacting the nutrient uptake and translocation of “nontarget” plants more than Fe and Mn [36], reducing weed control between 43 and 59 percent if done in the same solution before application, reducing the biomass of Zn by 88 to 96 percent when applied alone, or reducing 41 percent of the biomass of Zn in coapplication [37]. Foliar absorption is too greatly reduced when glyphosate is used in solutions with Zn, Ca, Fe, Mg, and Mn [38]. Comparing N, P, K, Cu, and Zn, N and Zn foliar nutrient levels decreased the most during observed postglyphosate application, and both were correlated with glyphosate concentration [39,40]. A sunflower reduces the absorption and translocation of radiolabeled Fe^59^, Zn^65^, and Mn^54^ [36,41]. Plants intoxicated by glyphosate present the same morphological alterations as those observed for N, B, Fe, and Zn deficiency [42,43,44,45]. These reports corroborate with our findings and indicate that Zn’s intense interaction with glyphosate may be a problem for glyphosate detection in soils. It remains to be demonstrated whether ZnO NCs would have a stronger binding affinity than the Zn ions found in soil.

After interaction with a nanocrystal, the vibrational modes of some glyphosate structures were diminished, indicating that they lost degrees of freedom. In contrast, others showed an increased intensity, probably indicating freer functional groups. The C=O of free COOH and C=O of H bonded COOH chemical bonds has the lowest intensity, followed by the NH_2_ and P-O of PO_3_H, neither of which are strong interactions with nanomaterials. When GLIFOTAL TR did not interact with nanocrystals, its vibrational status was low (1732 cm^−1^), but it increased by 1% when interacting with ZnO NCs.

These data suggest that while Ag fully interacts with glyphosate, ZnO NCs do not interact in corresponding 1732 cm^−1^, 1720 cm^−1^, and 1268 cm^−1^ wavenumber regions. Stronger interactions were related to bands at 1568 cm^−1^, 1557 cm^−1^, 1170 cm^−1^, 1081 cm^−1^, and 1031 cm^−1^. Pearson’s correlation (R) showed that interactions containing Ag are stronger than reactions with ZnO NCs (see Figure 4). These results were confirmed in AFM images, especially because they showed a higher interaction of Ag with glyphosate. On the other hand, glyphosate molecules were aggregated around ZnO. In both cases, the NCs aggregated not only glyphosate but also everything that composes GLIFOTAL TR. This interaction is strong enough to retrieve all nanocrystals, water, surfactants, or any other material at the mica sheet’s surface.

The band at 1170 cm^−1^ is attributed to R-PO(OH)_2_. This group can form strong complexes with metal that may result in adsorption, photodegradation, biodegradation processes, and the formation of soluble and insoluble complexes [18]. Therefore, based on FTIR results, the phosphate groups of the glyphosate molecules were binding on the surface of ZnO NCs.

It is reported in the literature that the use of ZnO NCs placement on the surface of multiwalled carbon nanotubes can enhance electrochemiluminescence signals to detect glyphosate [46]. Detection could reach limits lower than 1 μmol L^−1^. Alongside this, these methods are cheaper, faster, and more sensitive than many spectroscopic tests. A sensor for detecting pesticides in water using the ZnCdSe quantum dot’s photoluminescence intensity has also been developed [47]. However, to our knowledge, our investigation is the first to report the use of ZnO NCs as an FTIR enhancer to improve glyphosate detection. These results demonstrate that glyphosate infrared modes could be intensified due to the interaction between glyphosate and nanocrystals. The signal is likely reinforced due to a preconcentration of glyphosate molecules around of nanocrystal.

There are accurate and sensitive technologies related to atomic absorption spectrometry, electrothermal atomization atomic absorption spectrometry [48,49], flame atomic absorption spectrometry [50,51,52], and fading spectrophotometry. All of them suffer due to their requirement for well-established laboratory settings, their high complexity, and extended testing times. These spectrophotometric techniques could have their minimum limit of detection enhanced by ZnO NCs.

On the other hand, glyphosate’s ability to interact with ZnO NCs is changed in the presence of Ag (as seen with samples of Ag-doped ZnO and nanocomposite). We suggest that Ag interacts with phosphonate and carboxylate groups, resulting in signal suppression. Comparing our results with those of the PVP-capped silver nanocubes system for removing glyphosate from water [53], we corroborated those results in many aspects. It is possible that our nanomaterials could also be efficiently used for glyphosate removal from water. The difference between our work and the latter one is that while the latter has shown different absorption peaks for glyphosate FTIR spectra for PVP-capped silver nanocubes binding in a chemical process, our nanomaterials were relatively stable and homogeneous, suggesting that our NCs efficiently immobilized glyphosate in a physical process, since there was no change in the IR bands.

Ag_2_O NCs are separated from the primary Ag_2_O NCs aggregates, which are then surrounded by glyphosate. For this reason, we believe that Ag_2_O NCs could also be used to promote the removal of glyphosate from water. A photoluminescence study [53] has previously shown that silver nanocubes degrade the glyphosate present in drinking water. The study confirmed that silver nanoparticles could fully degrade glyphosate, evidenced by the intensity of photoluminescence spectra that gradually decreased as the glyphosate concentration diminished [54], concluding that the adsorption of the amine group of glyphosate is bound onto silver, leading to the aggregation of Ag_2_O NCs.

The characterization of glyphosate removal is not simple, mainly because it has never been found in a noncomplexed form in nonlaboratory situations. This means that it needs to be unbound first. In this context, the interaction between glyphosate and other metals is pH-dependent [55,56,57,58,59,60,61]. Glyphosate exists in a zwitterion form, with adjuvants or surfactants to improve its activity. It is an amino phosphonic analog of the natural amino acid glycine and can be protonated, presenting different ionic states depending on pH. Like an amino acid, it has a carboxylic acid and an amino group; the phosphonic acid can be ionized, while the second can be protonated [62]. Glyphosate assumes the following protonation states: pKa1 = 0.47, pKa2 = 5.69, and pKa3 = 11.81 [63], whereas at pH 6.0, both the phosphonate and the amino group are protonated [64], suggesting that strategies for glyphosate separation or purification based on pH changes are possible.

Finally, glyphosate has more than 1000 analogs [65]. Still, there are only two very similar analogs, which are useful to the same extent as glyphosate (N-hydroxy-glyphosate and N-amino-glyphosate) [66]. Once the phosphonates strongly adsorb onto almost all mineral surfaces [67], we believe that the enhancing properties of ZnO NCs could also be applied to them. On the other hand, we do not expect the same for amino-carboxylates, for which the interactions with mineral surfaces are weak, especially at a near-neutral pH [67]. Although the present data indicate that ZnO NCs can be used as an FTIR enhancer, enabling glyphosate detection for various technologies, this is a proof-of-concept study in the context of environmental water. Therefore, further studies are required to validate this glyphosate detection in agriculture and human/animal toxicology fields due to potential metabolization and the need to analyze the effect of soil and biofluids resulting from the acquired spectra. In this context, denoising and artificial intelligence algorithms can be applied to real-time detection, improving occupational safety.

## 5. Conclusions

In this work, we demonstrated for the first time that ZnO NCs may be used as enhancing agents for glyphosate detection using FTIR spectroscopy. Nanomaterials with Ag altered the ability of glyphosate to interact with ZnO NCs because Ag ions interact with phosphate and carboxyl groups, resulting in signal suppression. Thus, ZnO NCs are exciting tools for the signal amplification of GLIFOTAL TR, and nanomaterials with Ag (Ag-doped ZnO, nanocomposite, and Ag_2_O) could also be used to promote the removal of glyphosate from water. Therefore, we have demonstrated several advantages of ZnO NCs and nanocomposites.

## Figures and Tables

**Figure 1 nanomaterials-11-00509-f001:**
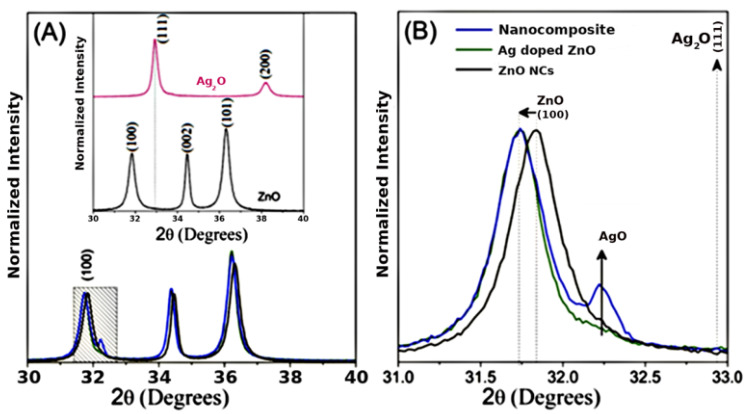
(**A**) X-ray diffraction (XRD) patterns of nanocomposite (blue line), zinc oxide (ZnO) (black line), Ag-doped ZnO (green line), and silver oxide (Ag_2_O) (pink line) nanocrystals. (**B**) The inset shows a magnification around the (100) diffraction plane.

**Figure 2 nanomaterials-11-00509-f002:**
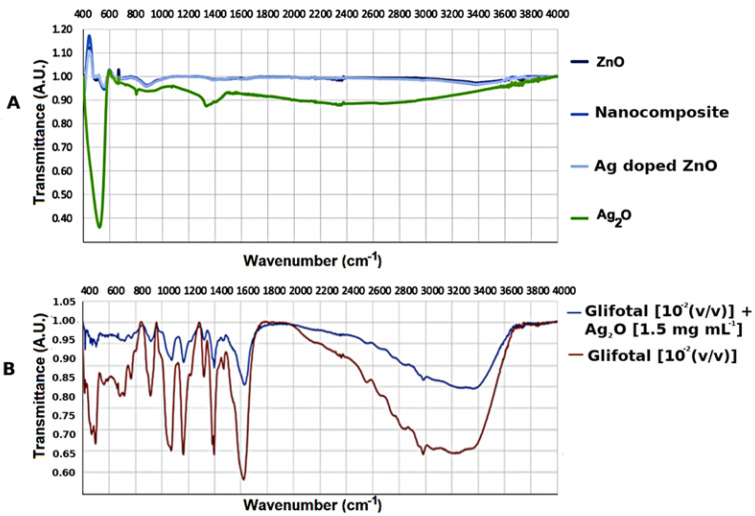
(**A**) Fourier transform infrared spectroscopy (FTIR) spectra of ZnO nanocrystals (NCs), Ag-doped ZnO, nanocomposite, and Ag_2_O NCs. (**B**) FTIR spectra of commercial formulation of glyphosate (GLIFOTAL TR^)^ and GLIFOTAL TR Ag_2_O NCs at 10^−2^ (*v*/*v*).

**Figure 3 nanomaterials-11-00509-f003:**
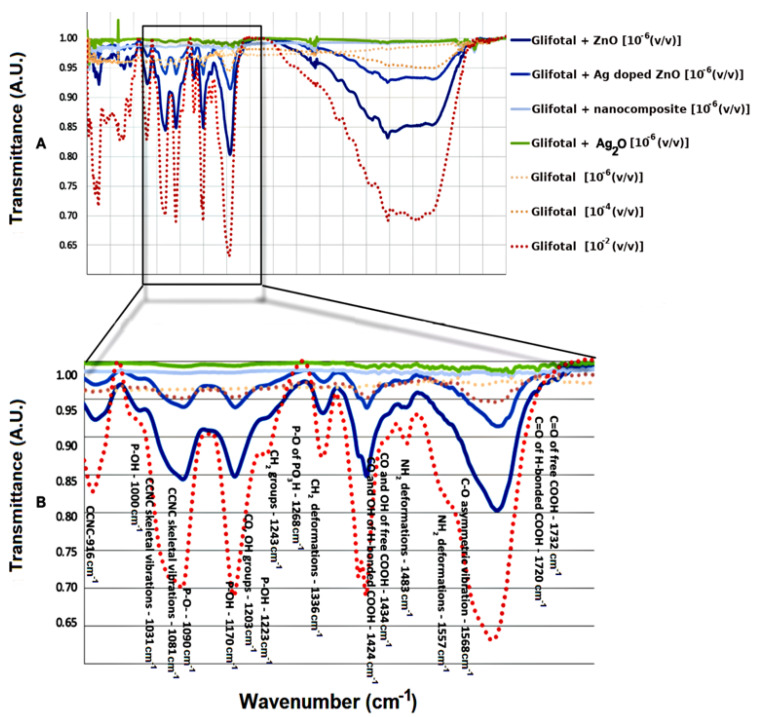
(**A**) FTIR spectra of GLIFOTAL TR at 10^−2^, 10^−4^, and 10^−6^ (*v*/*v*) and GLIFOTAL TR/ NCs at 10^−6^ (*v*/*v*). (**B**) Spectrum amplification.

**Figure 4 nanomaterials-11-00509-f004:**
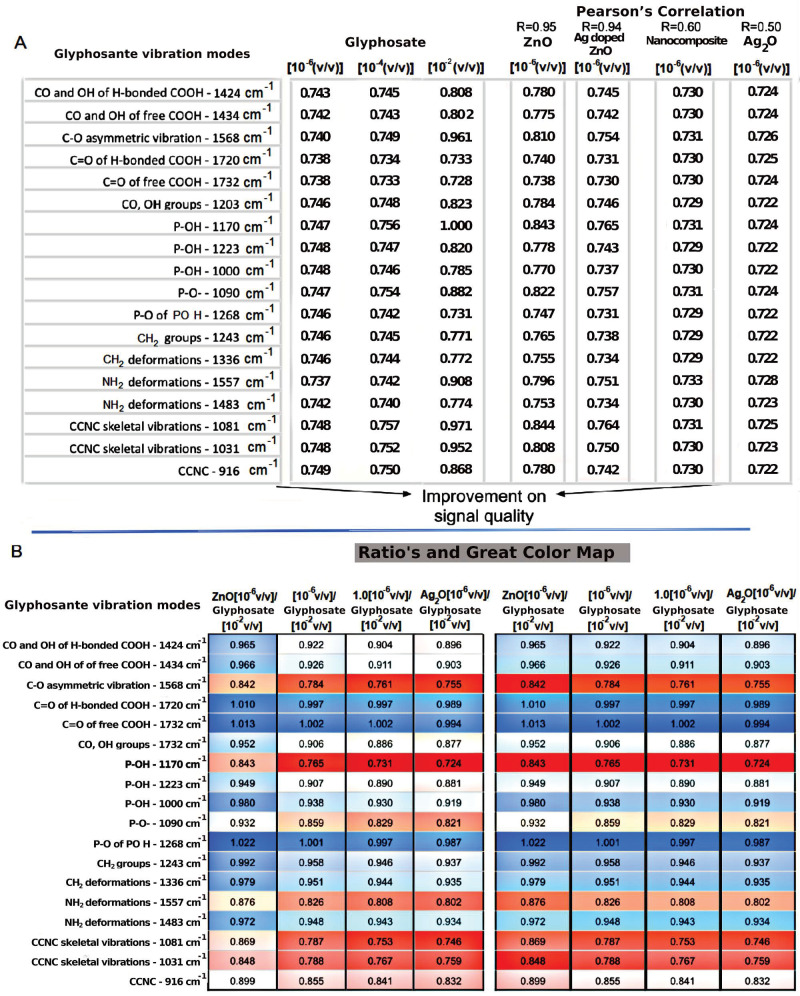
(**A**) The frequency mode of each molecular group that makes up glyphosate before and after interacting with nanomaterials. (**B**) An energy color scale related to frequency intensity changes when interacting with nanomaterials (ZnO, Ag_2_O, and Ag-doped ZnO (Ag ZnO)), and nanocomposite (ZnO AgO).

**Figure 5 nanomaterials-11-00509-f005:**
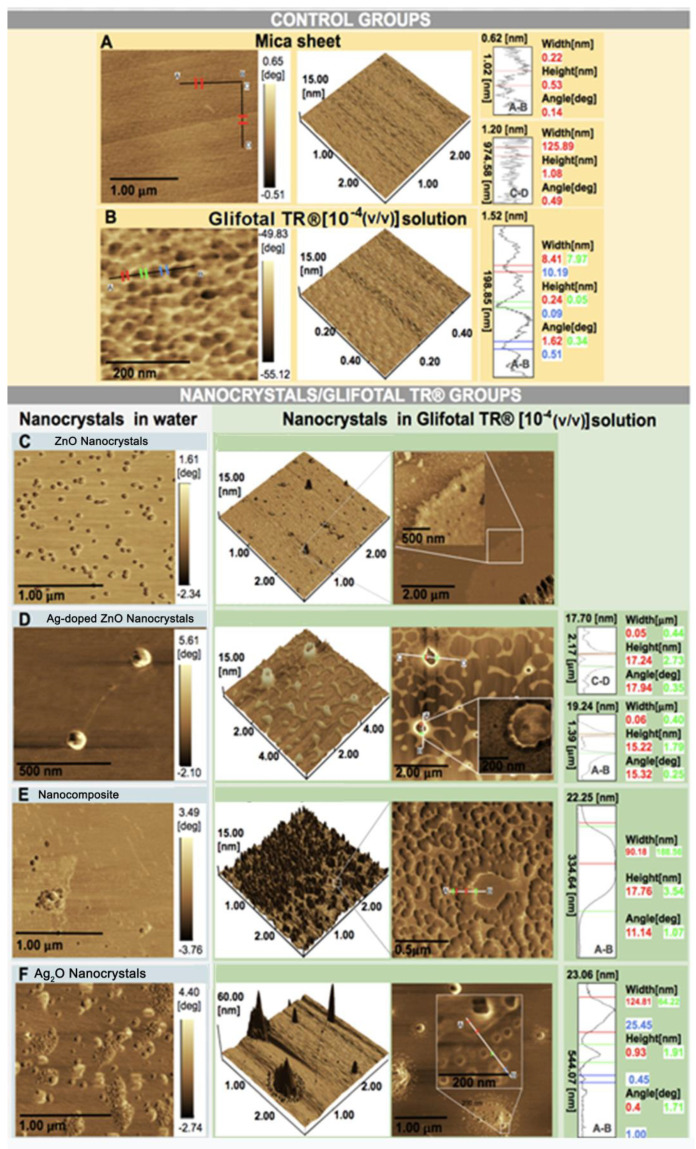
Atomic force microscopy (AFM) images of the control groups (mica and GLIFOTAL TR) and the nanocrystal/GLIFOTAL TR groups (ZnO, Ag-doped ZnO, nanocomposite, and Ag_2_O) in 10^−4^ (*v*/*v*).

## Data Availability

Not applicable.

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
