# Peer review of "Application of ZnO Nanocrystals as a Surface-Enhancer FTIR for Glyphosate Detection"

_nanomaterials, 2021, doi:10.3390/nano11020509_

Round 1

Reviewer 1 Report

“Application of ZnO Nanocrystals as a Surface2 Enhancer FTIR for Glyphosate Detection”

Let me start by declaring no potential or perceived competing interests that may influence my review.

Glyphosate is an effective broad-spectrum systemic herbicide, used successfully worldwide to control of unwanted weeds in crops and gardens. Since the expiration of its patent and the development of genetically modified crops resistant to the effects of glyphosate, this glyphosate has been widely used in different formulations. In fact, glyphosate can be found in a ready-to-use diluted formulation containing adjuvants and surfactants, as well as in concentrations as high as 48% v/v that require correct dilution before application. These formulations increase both the inherent risk of glyphosate to human health and the difficulty of its detection. In addition to the permitted concentrations of glyphosate, its detection requires expensive equipment and laborious processes. Nanotechnology has emerged as an alternative means for the detection of environmental pollutants due to its enormous potential for general cost reduction, reduced sample processing, as well as increased sensitivity. Therefore, the authors aimed to analyze the interaction of nanocrystals ZnO, Ag-doped ZnO, and Ag2O, as well as the nanocomposite (Ag-doped ZnO/AgO) with glyphosate and possible implications in biosensing applications by Fourier transform infrared spectroscopy, and atomic force microscopy.

Minor concerns:

Although the manuscript is grammatically well written, it is too long and does not explain how this methodology may be cost-effective and more sensitive in relation to others already in use.

Please clarify

Line 101: “Two 2 (µL)”

Line 106: “Three (3) µL”

The authors mean that only 2 or 3 µL were used or mean that 2 or 3 µL were used twice or more?

Line 165: The authors mean: In Figure 4(A)?

Line 139: Instead of “environment temperature” use 2room temperature” as previously.

Line 386: Reference’s year in bold, please

Correct reference 36

Major concerns:

The introduction is too long.

The limitations of the study should also be clearly indicated.

Glyphosate is absorbed mainly through the leaves of plants and then transported by the plant, inhibiting the action of enzymes responsible for the synthesis of amino acids. This metabolic pathway in plants is absent in animals, justifying the low toxicity attributed to glyphosate, however, some studies have shown that glyphosate can also affect enzymes present in animals. How can the authors explain how their methodology may overcome this metabolization?

Methods for glyphosate detection require expensive equipment and laborious processes. But Fourier transform infrared spectroscopy and atomic force microscopy are also expensive equipment, are not available in any laboratory, and need skilled labour. Therefore, how could the present methodology be cost-effective in daily use for glyphosate detection?

And when the current methodology is evaluated to detect glyphosate in real water samples? What will happen? This must be done before acceptance of the manuscript for publication.

The above manuscript should also be discussed:

Pan S, Chen X, Li X, Jin M.J Sep Sci. 2019 Mar;42(5):1045-1050. doi: 10.1002/jssc.201800957. (this manuscript is not authored by the reviewer)

The manuscript has so many self-citations, some of which are no longer appropriate, please remove them.

Author Response

We thank Reviewer #1 for the careful reading of our manuscript and providing valuable and insightful comments.

Answer: # Thank you for your considerations. We also believe that the presented data provide a comprehensive review and update on the application of ZnO Crystals as a surface enhancer FTIR for Glyphosate detection, which may hold the keys to open new avenues for fruitful discussions and insight to protect the environment for future generations. As requested, we revised the English language.

Minor concerns:

Although the manuscript is grammatically well written, it is too long and does not explain how this methodology may be cost-effective and more sensitive in relation to others already in use.

Please clarify

Line 101: “Two 2 (µL)”

Line 106: “Three (3) µL”

The authors mean that only 2 or 3 µL were used or mean that 2 or 3 µL were used twice or more?

Answer: Thanks for the comment. We rewrote the sentence to facilitate understanding.

“The samples were dropped 2 µL, twice, and dried using a triple dental syringe to obtain FTIR spectra.”

Line 165: The authors mean: In Figure 4(A)?

Answer: We appreciate the observation. So, we add (A) at the beginning of the caption.

Line 139: Instead of “environment temperature” use room temperature” as previously.

Answer: We appreciate the observation and perform the correction.

Line 386: Reference’s year in bold, please

Correct reference 36

Answer: We appreciate the observation and perform the correction.

______________________________________________________________________

Major concerns:

  • The introduction is too long.

Answer:  We understand your concern. As requested, we reduced the introduction.

  • The limitations of the study should also be clearly indicated.

Answer: Thank you. As requested, we inserted a sentence with the limitation of the study as follows: “Although the present data indicates that ZnO NCs can be used as an FTIR enhancer enabling technology for glyphosate detection, this is a proof-of-concept study in the context of environmental water. Therefore, further studies are required to validate this glyphosate detection in agriculture and human/animal toxicology fields due to the potential metabolization and the need to analyze the effect of soil and biofluids from the acquired spectrum. In this context, de-noising and artificial intelligence algorithms can be applied to real-time detection, improving occupational safety.”

  • Glyphosate is absorbed mainly through the leaves of plants and then transported by the plant, inhibiting the action of enzymes responsible for the synthesis of amino acids. This metabolic pathway in plants is absent in animals, justifying the low toxicity attributed to glyphosate, however, some studies have shown that glyphosate can also affect enzymes present in animals. How can the authors explain how their methodology may overcome this metabolization?

Answer: We understand your concern. In general, most functional groups of each herbicide are maintained during the metabolization in the biological field. So, it is expected the maintenance of these vibrational groups in the spectra relative to glyphosate. Considering that some vibrational modes can be changed during the metabolization, we inserted your concern as a limitation of the study. Although further studies are needed to clarify this issue, we have worked with several de-noising and artificial intelligence algorithms to improve the detection of specific molecules which can overcome this metabolization.  

  • Methods for glyphosate detection require expensive equipment and laborious processes. But Fourier transform infrared spectroscopy and atomic force microscopy are also expensive equipment, are not available in any laboratory, and need skilled labour. Therefore, how could the present methodology be cost-effective in daily use for glyphosate detection?

Answer: It is important point out that atomic force microscopy was only used to support the interaction between nanocrystals (NCs) and glyphosate. Although atomic force microscopy is a state-of-the-art technology, we do suggest that this tool could be applied to substitute the current tools used for glyphosate detection. From an economic perspective, some ATR-FTIR platforms are expensive; however, considering the very low-cost to perform the infrared analysis, this simple, non-destructive, sensitive, and highly reproducible physicochemical analytical technique is a potential cost-effective test for glyphosate detection. Besides, this protocol using  ATR-FTIR technology reduces the use of hazardous elements to glyphosate detection using the current techniques. Several manufacturing companies have developed portable ATR-FTIR platforms  to offer permanently high quality at the most competitive prices on the market. 

  • And when the current methodology is evaluated to detect glyphosate in real water samples? What will happen? This must be done before acceptance of the manuscript for publication.

The above manuscript should also be discussed:

Pan S, Chen X, Li X, Jin M.J Sep Sci. 2019 Mar;42(5):1045-1050. doi: 10.1002/jssc.201800957. (this manuscript is not authored by the reviewer)

Answer: We understand your concern. Our study presents a pivotal step in translating ATR-FTIR spectroscopy into glyphosate detection in water and agriculture policy scenarios. This step towards high-sensitive glyphosate analysis using a cost-effective protocol and simple processes has implications in the field of IR spectroscopy and environmental challenges. Analysis of water and soil using this technique would fit ideally in the environmental challenges as a glyphosate detection tool. Besides, the reduced time to detect glyphosate in biofluids of infected subjects could be pivotal to managing treatment while also bringing cost benefits to the health services. However, glyphosate analysis in soil, river waters, and/or biofluids will be further analyzed together with de-noising and artificial intelligence algorithms.

  • The manuscript has so many self-citations, some of which are no longer appropriate, please remove them.

Answer: As requested, we revised the citations in the manuscript.

Reviewer 2 Report

In this work, ZnO nanocrystals were used to enhance the FTIR signal of glyphosate, thus making its detection (which is generally tricky), somewhat easier. This work has some interest, but the authors need to address several issues before considering publication:

Page 2, line 88: If the methodology of synthesis is patented, this means that we can’t have access to information used for the synthesis of the particles? Such as precursors, ligands, synthesis temperature, solvent… Isn't that a bit strange?

Line 123: If these sizes are found by Scherrer equation, you need to mention this. Because the XRD-derived crystalline grain sizes may be different from those found by TEM or AFM, for example.

In some of the last paragraphs of the Introduction, the way that you introduce Ag and its oxide is not the best one. OK, you use it and you use ZnO, too. But try to introduce Ag in a smooth way. Otherwise it seems that it comes ‘too suddenly’, without connections with previous parts of the text.

Line 132: If the main AgO peak according to the reference pattern that you use is located at around 32.3 deg, then the corresponding lattice plane with the highest intensity should be the (-1 1 1), not the (110) that you write at Figure 1B. Please check well the reference pattern that you use. I checked it in my XRD database.

Line 134: How do you find that the percentage of AgO is 22.4%? You need to explain how you found it! Also, is it in weight percentage or not? Be clear, be specific, please.

Line 135: Which interactions? Be specific.

Line 136: You write two times ‘in ultrapure water’ at the same line. Avoid repetitions.

Line 154: Where do we see this? There are diluted samples in the Figures of Page 2 in (10 -4) v/v, 10 -6 or 10 -2 (v/v). But not ’10 -3’ (v/v)’ as you mention. So, why and how do you write that comparison? It is a mistake.

Figures 4 and 5 do not have high resolution.

Line 230: How do those reports ‘corroborate with your findings’? In which sense? You show that your ZnO Nps are ‘beneficial’, right? So, how those reports corroborate?

Line 243: Where are these volcanic structures? In which image of the Figure 5, exactly? Be more specific.

Are the AFM images able to confirm the size determined by XRD? Are AFM and XRD-derived size values similar?

Line 249: So, there is some kind of ‘alignment’ of glyphosate molecules around ZnO NCs, you mention. At line 95 you write that you simply mix GLIFOLAT with the ZnO. So, how is this mixing done? Shouldn’t it be done in a specific, clearly explained (and done) way in order to ensure that the glyphosate-ZnO interaction takes place in the best possible way? For example, how long does the mixing take? Is it with stirring, fast, medium speed, slow? Be please more specific, I repeat.

Line 265: This ‘organizational interaction’ can be maybe written more clearly? What is this about?

The Conclusion section has to be expanded and improved. It is not so clear in the manuscript what did Ag (or Ag oxides) offer finally to this application? Was their role only detrimental or also beneficial in some extent, in something?

Reference 12: Something is missing there. Page number probably. You write 2016, 15. And? If 15 is the volume number, page is missing.

Ref 14: Similarly, something is missing, probably page number.

Check also Ref 22 for the same comment. In Ref 22, year is 2010, volume is probably 22, page is what?

Check also Reference 31. For the same reason.

Check Ref 33, same reason. Page (and volume) numbers, where are they?

Check if other works involving ZnO and glyphosate merit citation, such as: Biocatalysis and Agricultural Biotechnology, Vol 22, year 2019, page 101434.

Author Response

We thank Reviewer #2 for careful reading of our manuscript and providing valuable and insightful comments.

  • Page 2, line 88: If the methodology of synthesis is patented, this means that we can’t have access to information used for the synthesis of the particles? Such as precursors, ligands, synthesis temperature, solvent… Isn't that a bit strange?

Answer: Thanks for the comment. The patent of the samples is not related to this experiment, but to parameters of synthesis that can be obtained to obtain interesting biological properties. The methodology adopted was via coprecipitation in the absence of surfactants, so all glyphosate interactions occur directly with the nanoparticles. So, we added these details in the Materials and Methods section.

______________________________________________________________________

  • Line 123: If these sizes are found by Scherrer equation, you need to mention this. Because the XRD-derived crystalline grain sizes may be different from those found by TEM or AFM, for example.

Answer: We appreciate that observation. The difference in values between those obtained by XRD and those of AFM images is related to crystallinity. In XRD results the size is that of the crystallite and not necessarily the nanoparticle. In AFM images, the size obtained is associated with a nanoparticle. Thus, in order to avoid doubts, we remove the sentence that we comment on the values in the XRD results.

______________________________________________________________________

  • In some of the last paragraphs of the Introduction, the way that you introduce Ag and its oxide is not the best one. OK, you use it and you use ZnO, too. But try to introduce Ag in a smooth way. Otherwise it seems that it comes ‘too suddenly’, without connections with previous parts of the text.

Answer: Thanks for the comment and perform the correction.

______________________________________________________________________

  • Line 132: If the main AgO peak according to the reference pattern that you use is located at around 32.3 deg, then the corresponding lattice plane with the highest intensity should be the (-1 1 1), not the (110) that you write at Figure 1B. Please check well the reference pattern that you use. I checked it in my XRD database.

Answer: We appreciate that observation and perform the correction.

______________________________________________________________________

  • Line 134: How do you find that the percentage of AgO is 22.4%? You need to explain how you found it! Also, is it in weight percentage or not? Be clear, be specific, please.

Answer: Thanks for the comment and perform the correction.

______________________________________________________________________

  • Line 135: Which interactions? Be specific.

Answer: Thanks for the comment. We rewrote the sentence to specify the interactions.

“In order to investigate the interactions of nanomaterials with glyphosate functional groups, the infrared spectroscopy technique was used.”

______________________________________________________________________

  • Line 136: You write two times ‘in ultrapure water’ at the same line. Avoid repetitions.

Answer Thanks for the comment and perform the correction.

______________________________________________________________________

  • Line 154: Where do we see this? There are diluted samples in the Figures of Page 2 in (10 -4) v/v, 10 -6or 10 -2 (v/v). But not ’10 -3’ (v/v)’ as you mention. So, why and how do you write that comparison? It is a mistake.

Answer: Thanks for the comment and perform the correction.

______________________________________________________________________

  • Figures 4 and 5 do not have high resolution.

Answer Thanks for the comment and perform the correction.

______________________________________________________________________

  • Line 230: How do those reports ‘corroborate with your findings’? In which sense? You show that your ZnO Nps are ‘beneficial’, right? So, how those reports corroborate?

Answer: Thanks for the comment. In other studies we have demonstrated that these nanomaterials are biocompatible, this is important for this work, as it is an important parameter in applications. We believe that these results are important for generating devices for detecting glyphosate using ZnO. Being a simple method of just mixing and detecting.

______________________________________________________________________

  • Line 243: Where are these volcanic structures? In which image of the Figure 5, exactly? Be more specific.

Answer Thanks for the comment. The volcanic structures are generated due to a strong interaction between the glyphosate and the ions of Ag. But we removed this section and left only the comment of aggregates.

______________________________________________________________________

  • Are the AFM images able to confirm the size determined by XRD? Are AFM and XRD-derived size values similar?

Answer: We appreciate that observation. As we mentioned earlier, the values of the results of XRD and AFM are a little different, so in order to avoid confusion, we remove those obtained by XRD. In addition, it is observed that the size in the nanoparticles are similar.

______________________________________________________________________

  • Line 249: So, there is some kind of ‘alignment’ of glyphosate molecules around ZnO NCs, you mention. At line 95 you write that you simply mix GLIFOLAT with the ZnO. So, how is this mixing done? Shouldn’t it be done in a specific, clearly explained (and done) way in order to ensure that the glyphosate-ZnO interaction takes place in the best possible way? For example, how long does the mixing take? Is it with stirring, fast, medium speed, slow? Be please more specific, I repeat.

Answer: Thanks for the comment. We rewrote this excerpt to facilitate understanding. Regarding the mixture made under medium pressure. We performed some tests to see if it affected, but there was no difference.

______________________________________________________________________

  • Line 265: This ‘organizational interaction’ can be maybe written more clearly? What is this about?

Answer Thanks for the comment and perform the correction. We comment that as the Ag ions have a strong interaction with glyphosate, there is a suppression of the modes. In the ZnO, however, as the interaction is not so strong, it does not suppress the modes.

______________________________________________________________________

  • The Conclusion section has to be expanded and improved. It is not so clear in the manuscript what did Ag (or Ag oxides) offer finally to this application? Was their role only detrimental or also beneficial in some extent, in something?

Answer: Thanks for the comment. We rewrote the Conclusion.

______________________________________________________________________

  • Reference 12: Something is missing there. Page number probably. You write 2016, 15. And? If 15 is the volume number, page is missing.

Ref 14: Similarly, something is missing, probably page number. Check also Ref 22 for the same comment. In Ref 22, year is 2010, volume is probably 22, page is what?

Check also Reference 31. For the same reason. Check Ref 33, same reason. Page (and volume) numbers, where are they?

Check if other works involving ZnO and glyphosate merit citation, such as: Biocatalysis and Agricultural Biotechnology, Vol 22, year 2019, page 101434.

       Answer Thanks for the comment and perform the correction.

Round 2

Reviewer 2 Report

The authors need to further improve their work, since the response to reviewers was only partially successful.

The authors still do not mention how they found that the AgO weight percentage in the nanocomposite is 22.4%.

Line 113: You write two times 'in ultrapure water' in the same phrase.

I had a comment, at previous version was line 154, now are lines 166-167. I don't think you addressed this comment: There are diluted samples in the Figures of Page 2 in (10-4) v/v, 10-6 or 10-2 (v/v). But not ’10-3’ (v/v)’ as you mention in the text. So, why and how do you write that comparison? It is a mistake.

Regarding my comment for the alignment (line 249 of previous version) and mixing (line 95 of previous version). You write that at the revised version you rewrote this excerpt. Where? Why don't you put yellow lines in your text to see these changes? And why you don't write to the reviewer exactly your answer? Please don't complicate the work of reviewers.

Author Response

Reviewer #2

Comments and Suggestions for Authors

The authors need to further improve their work, since the response to reviewers was only partially successful.

We thank Reviewer #2 for careful reading of our manuscript and providing valuable and insightful comments.

  • The authors still do not mention how they found that the AgO weight percentage in thenanocomposite is 22.4%.

Answer: Thanks for the comment. The phase percentage was performed by the reason of the integrated intensities, to have an estimate of the percentage of phase formed. Thus, we include this detail in the Materials and Methods section (Page 4, line 64).

______________________________________________________________________

  • Line 113: You write two times 'in ultrapure water' in the same phrase.

Answer: Thanks for the comment and perform the correction (Page 5 line 69 and Page 6 line 89).

______________________________________________________________________

  • I had a comment, at previous version was line 154, now are lines 166-167. I don't think you addressed this comment: There are diluted samples in the Figures of Page 2 in (10-4) v/v, 10-6 or 10-2 (v/v). But not ’10-3’ (v/v)’ as you mention in the text. So, why and how do you write that comparison? It is a mistake.

Answer: We appreciate that observation and perform the correction (Page 9 line 138)

______________________________________________________________________

  • Regarding my comment for the alignment (line 249 of previous version) and mixing (line 95 of previous version). You write that at the revised version you rewrote this excerpt. Where? Why don't you put yellow lines in your text to see these changes? And why you don't write to the reviewer exactly your answer? Please don't complicate the work of reviewers.

Answer: In relation to the previous review, we had removed volcanic structures term (Page 10 line 169 and Page 15 line 238) and add the interactions sentence in Page 7 line 115.

In this review we rewrite the excerpts that we commented on alignment in the Page 15 (line 238 and 243) and Conclusion.

___________________________________________________________________

Round 3

Reviewer 2 Report

In this new revision, the authors tried to improve some things but unfortunately, again, they were not so careful. This comment refers mainly to the quality of the english used at the new parts with highlighted yellow colour. At line 63: only two lines of yellow colour, but the english is not good. Line 243: This part can also be improved, in what concerns the english used. Line 71: 'To proving'. Why? And a scientific comment: How the mixing was done? You did not answer this question. Was there any stirring, for example?

Author Response

Reviewer #2

In this new revision, the authors tried to improve some things but unfortunately, again, they were not so careful.

We thank Reviewer #2 for careful reading of our manuscript and providing valuable and insightful comments.

  • This comment refers mainly to the quality of the english used at the new parts with highlighted yellow colour.

Answer: Thanks for the comment. We carry out the English correction.

___________________________________________________________________

  • At line 63: only two lines of yellow colour, but the english is not good. Line 243: This part can also be improved, in what concerns the english used. Line 71: 'To proving'. Why?

Answer: We agree as a reviewer and carry out the English correction, and we withdraw the part of “To proving'”

___________________________________________________________________

  • And a scientific comment: How the mixing was done? You did not answer this question. Was there any stirring, for example?

Answer: Thanks for the comment. We have added to the following sentence:

“The final solution was stirred manually at a medium speed.” (Page 5, line 71)

___________________________________________________________________

  • The authors still do not mention how they found that the AgO weight percentage in the nanocomposite is 22.4%.

Answer: Thanks for the comment. About the phase percentage value, a Gaussian adjustment was made to the crystalline phases' main peaks and calculated to the area (Integrated Intensity), followed by dividing the integrated intensity value of AgO by ZnO.

“The AgO phase percentage was determined by the ratio of the integrated intensities of AgO/ZnO diffraction peaks from XRD results.” (Page 4, line 64)

___________________________________________________________________
